# Voxel-wise deep learning segmentation of hydroxyapatite and iodine in spectral photon-counting CT: A quantitative phantom study

**Nadine Francis**[1]*, **Mohamed L. Seghier**[2], **Nabil Maalej**[1], **Aamir Raja**[1]*

1 Department of Physics, Khalifa University of Science and Technology, Abu Dhabi, United Arab Emirates, 2 Department of Biomedical Engineering and Biotechnology, Khalifa University of Science and Technology, Abu Dhabi, United Arab Emirates

* Nadine.samy.francis@hotmail.com (NF); aamir.raja@ku.ac.ae (AR)

## Abstract

Accurate non-invasive identification of hydroxyapatite (HA) deposits is important for diagnosing calcific musculoskeletal disease and quantifying vascular calcification, but conventional and dual-energy CT often struggle to distinguish HA from iodinated contrast because of overlapping attenuation, noise, and beam-hardening artifacts. Spectral photon-counting CT (SPCCT) offers improved energy resolution and spatial fidelity, yet most deep-learning approaches in spectral CT focus on continuous density regression or anatomical segmentation rather than direct voxel-wise material labeling. We developed SPFF–UNet, a spectral-preserving 3D segmentation model for direct classification of HA and iodine concentrations from five-bin SPCCT volumes without material-decomposition preprocessing. A cylindrical phantom containing twelve materials was scanned at 0.1 mm isotropic resolution, including five HA concentrations, three iodine concentrations, three soft-tissue equivalents, and water. SPFF–UNet integrates spectral squeeze-excitation, EnergyFiLM, and FourierGate to preserve and exploit multi-energy information throughout the network. The model was trained for thirteen-class voxel-wise segmentation and compared with five established 3D architectures under matched training conditions. SPFF–UNet achieved the best macro-averaged performance on a held-out phantom scan (Dice 0.72 ± 0.01, IoU 0.59 ± 0.01, sensitivity 0.73 ± 0.01, precision 0.71 ± 0.01), outperforming the strongest comparator, ResUNet++ (Dice 0.66 ± 0.02, IoU 0.46 ± 0.02, sensitivity 0.67 ± 0.02, precision 0.61 ± 0.03). Performance gains were concentrated in mid/low-contrast HA and low-concentration iodine, with reduced slice-wise variability and fewer HA–iodine misclassifications. These results suggest that preserving spectral information and applying targeted spectral modulation can improve concentration-aware voxel classification from SPCCT. This phantom-based proof-of-concept provides a basis for future in vivo validation.

**Data availability statement:** All relevant data will be provided upon request.

**Funding:** This project was funded by the Research and Innovation Grant, Khalifa University of Science and Technology, Abu Dhabi, UAE under account number 8474000563.

**Competing interests:** The authors have declared that no competing interests exist.

## Introduction

Hydroxyapatite (HA), a calcium phosphate that is the primary mineral component of bone and teeth, can abnormally deposit in soft tissues and blood vessels, contributing to a range of pathologies [1,2]. In the musculoskeletal system, HA accumulates in tendons and periarticular tissues and is implicated in conditions such as calcific periarthritis, osteoarthritis, and the destructive "Milwaukee shoulder" syndrome [3–5]. Beyond the musculoskeletal domain, HA deposition underlies vascular calcification, an early marker of arterial stiffness and cardiovascular risk. Coronary artery calcium visible on computed tomography (CT) images reflects HA burden and serves as a strong predictor of adverse events [2]. In oncology, clustered HA microcalcifications are frequently found in breast cancers and act as early diagnostic indicators, particularly in ductal carcinoma in situ [6]. Given its presence across musculoskeletal, cardiovascular, and oncologic conditions, HA acts as a distinctive imaging biomarker. Therefore, early and accurate detection of HA deposits is critical for timely diagnosis and treatment planning [3].

However, in vivo identification of HA remains challenging, particularly in distinguishing it from iodinated contrast on CT. Clinically, an important goal is to non-invasively classify HA at relevant concentrations and differentiate it from iodine, thereby reducing the need for aspiration or biopsy. Conventional confirmation of the presence and concentration of HA crystals relies on invasive sampling with crystal analysis. An imaging approach capable of concentration-aware HA labeling may ultimately help reduce procedural risk and broaden access to diagnosis, but this will require validation beyond phantom studies [3,5].

Conventional CT integrates photon counts across energies, effectively compressing the energy-dependent attenuation coefficient $\mu(E, Z, \rho)$ into a single CT number per voxel. As a result, low-concentration iodine (high atomic number, $Z$) and more densely mineralized HA (lower $Z$ but higher density) can yield similar effective attenuation. Intensity-based thresholds alone cannot reliably separate them, and misclassification is common, especially at low concentrations and in mixed voxels. Furthermore, dense calcifications generate beam hardening and blooming artifacts, which can obscure subtle or early HA deposits [7,8]. Dual-energy CT (DECT) improves material separation by acquiring two X-ray spectra and exploiting differences near the K-edge [9], and is widely used in clinical settings [10]. However, DECT material-separation accuracy remains limited due to overlapping spectral signatures, image noise, beam hardening, and limited spatial resolution [11,12]. Crosstalk between iodine and calcium (predominantly HA) is particularly common, while misregistration and photon starvation further degrade material separation and detectability; early HA deposits often remain below the detection limit [8,13–15]. These constraints underscore the need for advanced spectral imaging and analysis techniques that can accurately and reliably separate HA from iodine in vivo.

Spectral photon-counting CT (SPCCT) addresses key limitations of conventional and dual-energy CT [16]. Photon-counting detectors (PCDs) register individual X-ray photons and sort them into multiple energy bins, enabling simultaneous multi-material decomposition beyond the two-bin limit of DECT [9,17]. Since the

introduction of clinical SPCCT scanners, evaluations have progressed from technical feasibility to clinical studies across cardiovascular, thoracic, abdominal, musculoskeletal, neuro, and pediatric imaging [18]. SPCCT provides higher spatial resolution (voxel size ~0.1 mm), improved contrast-to-noise ratio, reduced beam hardening, and minimal electronic noise. These advances enhance imaging of calcifications (predominantly HA) and HA–iodine separation, as shown by others in coronary and other applications [9,11,19–21]. Phantom and in vivo studies further demonstrate improved decomposition of HA, iodine, and soft tissue, even in the presence of metallic implants [22]. Complementary to spectral hardware advances, deep learning-based CT metal artefact reduction methods have been proposed to improve image quality in the presence of implants [23]. Building on these hardware and reconstruction advances, deep learning (DL) methods are increasingly used to convert spectral data into clinically interpretable outputs, with early work supporting multi-material decomposition. Beyond material mapping, DL has also been applied to broader SPCCT workflows, such as automated brain segmentation for image quality and artifact assessment [24], coronary plaque quantification sensitive to temporal resolution [25], and diagnostic coronary artery disease detection in SPCCT angiography [26]. Collectively, these studies demonstrate feasibility, yet largely emphasize anatomical or disease-level tasks rather than concentration-aware, voxel-wise material classification.

DL can learn nonlinear mappings from energy-resolved CT data to material-specific outputs, often outperforming least-squares or iterative decompositions. Convolutional neural networks (CNNs) exploit spatial context, improving noise robustness and reducing dependence on calibration. Prior work in spectral CT largely follows two strategies: (i) continuous regression to predict material density maps, and (ii) discrete classification/segmentation to assign material labels. Most studies fall into the regression category. Encoder–decoder CNNs regress voxel-level densities for multiple materials and improve over physics-based methods [27]. For instance, Han et al. trained MD-UNet on three-bin SPCCT phantoms with HA (100, 300 mg/mL) and iodine (2–15 mg/mL), producing water, calcium, and iodine maps with lower error and higher CNR than conventional decomposition [8]. Similarly, Raja et al. systematically quantified identification and quantification errors in spectral CT material decomposition, underscoring the critical importance of accurate voxel-level evaluation in multi-material separation tasks [28]. Rajagopal et al. proposed a two-stage network—material presence detection followed by concentration regression (iodine, gadolinium, calcium)—achieving high accuracy for iodine/gadolinium but higher error for calcium [29]. While effective, regression typically requires post hoc thresholding to yield categorical labels. In this study, we favored classification over regression because the target outputs were predefined, clinically interpretable concentration categories rather than continuous material-density estimates. A classification framework yields direct voxel-wise labels (e.g., specific HA or iodine concentration classes) without requiring an additional post hoc discretization step. This is particularly suitable for the present phantom design, in which the rods were prepared at discrete known concentration levels. Regression remains important for continuous quantification tasks, but classification better matches the present objective of concentration-aware voxel labeling. In contrast, fewer studies pursue direct classification, which yields discrete, interpretable maps (e.g., "HA 50 mg/cm³" or "iodine 10 mg/mL") suitable for localization and quantification [30]. Sigurdsson et al. applied ResUNet++ to SPCCT basis images to generate iodine-only maps with calcium suppressed [31], but relied on pre-reconstructed basis images, targeted iodine (not HA), and lacked concentration-specific labels. Baek et al. trained 3D UNet and Swin UNETR on four-energy SPCCT for abdominal organ segmentation [32], an anatomy-focused task rather than concentration-aware HA–iodine classification. To our knowledge, no prior work trains a model for automated voxel-level labeling of HA and iodine directly from SPCCT data at clinically relevant concentrations, a capability aligned with the diagnostic objectives outlined above.

Here, we investigate whether discrete, concentration-aware voxel labels can be generated directly from raw SPCCT volumes without ROI selection or basis-image preprocessing, with the aim of improving separation of hydroxyapatite and iodine at clinically relevant concentrations. We introduce SPFF–UNet, a spectral-preserving 3D U-Net that preserves the energy axis end-to-end and integrates EnergyFiLM and FourierGate, and we benchmark it against five established 3D segmentation models under a shared training scheme, reporting mean±SD across multiple seeds. We also propose

grid-puzzle augmentation, which randomizes global layout while preserving local spectral texture, and we conduct comprehensive ablations isolating each component's contribution. While phantom-based, these discrete, concentration-aware maps are intended to support future in vivo evaluation.

## Materials and methods

Fig 1 summarizes the study workflow: (A) input data and dataset preparation, (B) the evaluated U-Net–based model variants, and (C) output generation and quantitative/qualitative evaluation. This study used phantom data only and did not involve human participants, animals, or identifiable patient data; therefore, ethics approval and informed consent were not required.

### SPCCT acquisition

Datasets were acquired on a MARS Microlab 5×120 SPCCT scanner [33–35] with twelve Medipix3RX PCDs bump-bonded to 2 mm CZT, forming a 16.8 cm × 1.4 cm array (110 μ m pixels). Up to eight energy bins are available; we used five in charge-summing mode to reduce charge sharing [36,37]. The arbitration counter was set at ~7 keV, and energy calibration used threshold scans of the Bremsstrahlung spectrum at multiple anode voltages [38,39]. The X-ray source

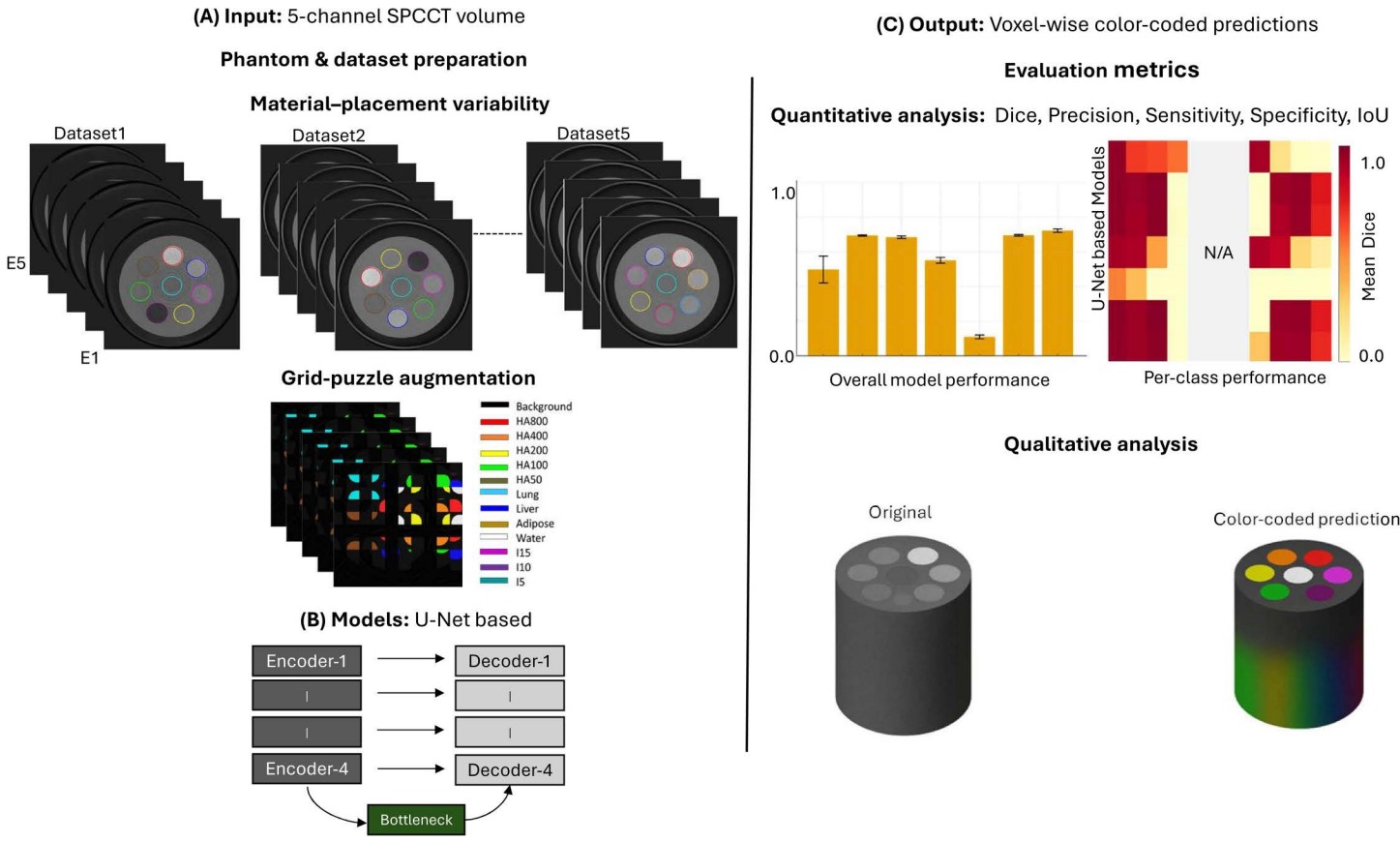

**Fig 1. Methodology overview. (A)** Input multi-energy SPCCT data and dataset preparation (including material/placement variability and grid-puzzle augmentation). **(B)** Deep learning models evaluated (six U-Net–based variants) under a unified training configuration. **(C)** Outputs and evaluation, including macro-averaged overall and per-class metrics (Dice, sensitivity, specificity, precision, IoU) and qualitative overlays / slice-wise Dice-error plots ($e = 1 − \text{Dice}$). Abbreviations: SPCCT, spectral photon-counting CT; IoU, intersection over union.

(SourceRay SB-120–350) provided a polyenergetic beam with 1.8 mm Al-equivalent intrinsic and 0.125 mm brass filtration. A low-energy protocol with five bins (7–12, 12–15, 15–18, 18–21, 21–120 keV) was selected via a grid-search optimization adapted from [40] to provide fine spectral resolution for distinguishing hydroxyapatite and related calcium crystal types. Volumes were reconstructed with the MARS iterative algorithm at 0.1 mm isotropic voxels on a 1300 × 1300 grid.

## Phantom and dataset preparation

Given limited clinical SPCCT data for HA crystal imaging, we used a custom cylindrical phantom (100 mm diameter) with eight insert positions (1 central, 7 in a ring) that held interchangeable material rods. Twelve materials were used: HA at 50/100/200/400/800 mg/cm³; iodine at 5/10/15 mg/mL; three soft-tissue equivalents (adipose, liver, lung; PTW, Freiburg, Germany); and water. Five independent scans were acquired, each with a different set of eight rods, so all materials are represented across the dataset. Rods (10 mm diameter) traversed the phantom and thus appear in every slice; each scan has 100 axial slices. Although not anthropomorphic, the 100 mm size approximates extremities/joints and yields realistic beam hardening while preserving experimental control. To decouple spectral cues from geometry, we apply grid-puzzle augmentation: for each multi-energy slice, sample $g \in \{1, \ldots, 10\}$, partition into a $g \times g$ grid, randomly permute tiles, and apply the same permutation to all energy bins and the label map. This preserves local spectral texture and voxel-level supervision while disrupting global shapes (rod contours, ring layout), encouraging reliance on energy-dependent attenuation rather than geometry priors. Using larger grids (e.g., $g = 10$) produces small tiles that emphasize crystal-scale patterns, ensuring exposure to small calcification sizes rather than only the large, fixed rod geometry. In training, grid-puzzle is attempted for every sample and applied when $g > 1$; it is disabled for validation and test. Additional augmentations: random flips, 90° rotations, mild brightness scaling, and additive Gaussian noise. Data splitting was performed at the scan level to avoid leakage between highly correlated slices from the same acquisition. Four scans were used for model development, with an 80/20 train/validation split, and a fifth entirely separate scan was held out for external testing. Fig 2 summarizes the datasets and augmentation. The dataset underlying this study is publicly available in IEEE DataPort (https://doi.org/10.21227/gbhn-nk95).

## Deep learning framework

We developed a voxel-level multiclass segmentation framework for SPCCT volumes with multi–energy bins per slice. Inputs are tensors $\mathbf{x} \in \mathbb{R}^{B \times 1 \times F \times H \times W}$, where $B$ is the batch size, $F$ the number of energy bins, and $(H, W)$ the in-plane resolution ($H = W = 512$); the second dimension is the channel axis, which is 1 at the input. The network predicts voxel-wise logits $\hat{\mathbf{y}} \in \mathbb{R}^{B \times C \times H \times W}$, where $C$ denotes the number of classes ($C = 13$; HA/iodine concentrations and soft-tissue equivalents).

We benchmarked five established 3D segmentation models alongside a customized spectral-preserving architecture tailored to SPCCT. We first summarize the baselines, then describe our proposed model. All models used a unified training protocol: identical splits, preprocessing, and augmentation (including grid–puzzle); the same batch size; and early stopping on validation macro Dice (mode: max) with patience 12 and minimum improvement $10^{-3}$. Training was capped at 200 epochs but typically stopped earlier via early stopping; validation was computed at each epoch on the held-out split. Each model was trained with three random seeds, and results are reported as mean±SD under the same protocol.

## Baseline models

We implemented five widely used 3D segmentation networks as baselines: 3D UNet [41], a four-level volumetric encoder–decoder with skip connections originally developed for 3D microscopy and now common in CT/MRI [42,43]; ResUNet++ (3D adaptation) [44], a residual UNet with squeeze-and-excitation (SE) skips and an atrous spatial pyramid pooling (ASPP) bottleneck; R2UNet (3D) [45], which adds recurrent residual refinements to a UNet backbone (we follow the 3D implementation of [46]); Swin UNETR [47], which uses a hierarchical Swin-Transformer encoder with shifted windows for multi-scale volumetric features; and UNETR [48], which pairs a Vision Transformer encoder with a U-shaped CNN

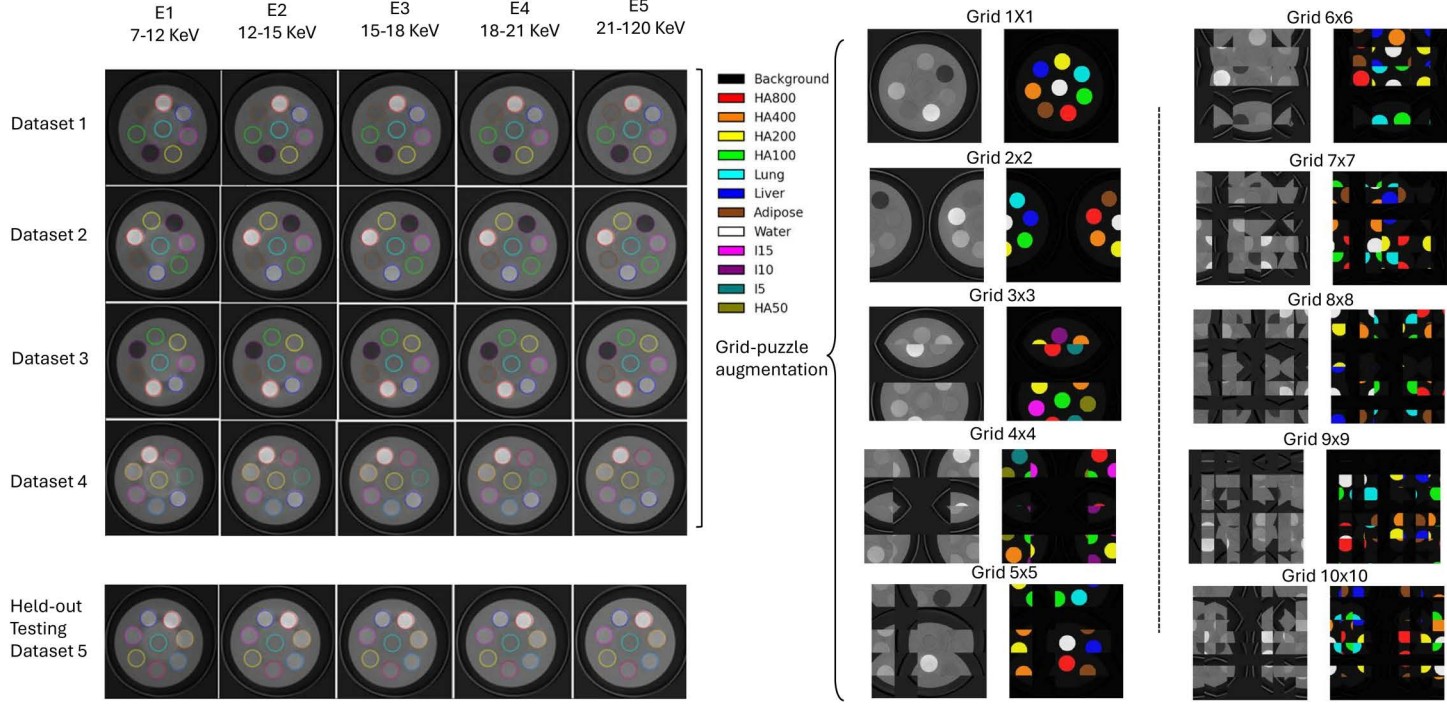

**Fig 2. Overview of the phantom datasets and augmentation.** Left: matrix view with scans as rows and energy bins ($E_1$ to $E_5$) as columns. Right: $n \times n$ grid puzzle augmentation on a representative slice; the original grayscale DICOM image and the corresponding ground truth (voxels color-coded by class) are shown.

decoder. Together, these baselines span classical CNNs (3D UNet), UNet augmentations (ResUNet++), recurrent refinement (R2UNet 3D), and transformer-based encoders (Swin UNETR, UNETR), allowing us to test whether recurrence and transformer/attention mechanisms tailored for volumetric data provide advantages over strong CNN baselines. Architectural and training settings follow the original sources; where hyperparameters were not reported, we adopted reasonable defaults. Table 1 summarizes final training setups for all baseline models.

## Proposed spectral-preserving model

We introduced the Spectral-Preserving Fourier-FiLM U-Net (SPFF–UNet), a spectral-preserving UNet hybrid tailored to SPCCT. The backbone is a 3D UNet with four encoder-decoder levels and skip connections; down/upsampling is anisotropic ($1 \times 2 \times 2$), so the energy axis $F$ is preserved end to end. On this backbone, two modules operate along the energy dimension to exploit multi-bin information. First, EnergyFiLM performs per-energy affine modulation: after each double-convolution block, features $\mathbf{u} \in \mathbb{R}^{B \times C_{\text{feat}} \times F \times H \times W}$ (where $C_{\text{feat}}$ is the number of feature channels) are modulated by parameters $(\gamma, \beta) \in \mathbb{R}^{B \times C_{\text{feat}} \times F}$ predicted from a sinusoidal encoding of the energy index and broadcast across ($H, W$). Second, FourierGate computes a global spectral profile (mean over channels and space), applies a real FFT (rFFT) along $F$, multiplies by a learnable magnitude mask, and inverse-transforms to obtain a gate $\mathbf{w} \in \mathbb{R}^{B \times 1 \times F \times 1 \times 1}$ that rescales $\mathbf{u}$. Together, EnergyFiLM and FourierGate ensure that SPFF–UNet not only preserves the $F$ energy bins, but also exploits energy-dependent attenuation trends and frequency-domain structure (e.g., smooth spectral variations or K-edge-like transitions) when producing voxel-wise predictions. A spectral squeeze–excitation (spec-SE) provides complementary channel reweighting. Fig 3 illustrates the overall architecture and module placement.

**Table 1. Final training setups for baseline models.** Hyperparameters reported in the original papers are retained; missing values for optimizer (Opt.), learning rate (LR), schedule (Sched.), momentum (Mom.), or weight decay (WD) were set to common defaults and marked with †. SGD: stochastic gradient descent; CE: cross-entropy; Dice: Dice loss. Warm+Cos = linear warm-up followed by cosine annealing of the learning rate.

| Model | Opt. | LR | Sched. | Mom. | WD | Loss |
|---|---|---|---|---|---|---|
| 3DUNet [41] | SGD | 1e-2† | – | 0.99† | – | CE |
| UNETR [48] | AdamW | 1e-4 | – | – | 1e-2† | Dice+CE |
| R2UNet3D [45] | Adam | 1e-3 | – | – | – | Dice |
| SwinUNETR [47] | AdamW† | 8e-4 | Warm+Cos | – | 1e-2† | Dice |
| ResUNet++ [44] | Adam | 1e-4 | – | – | – | Dice+CE |

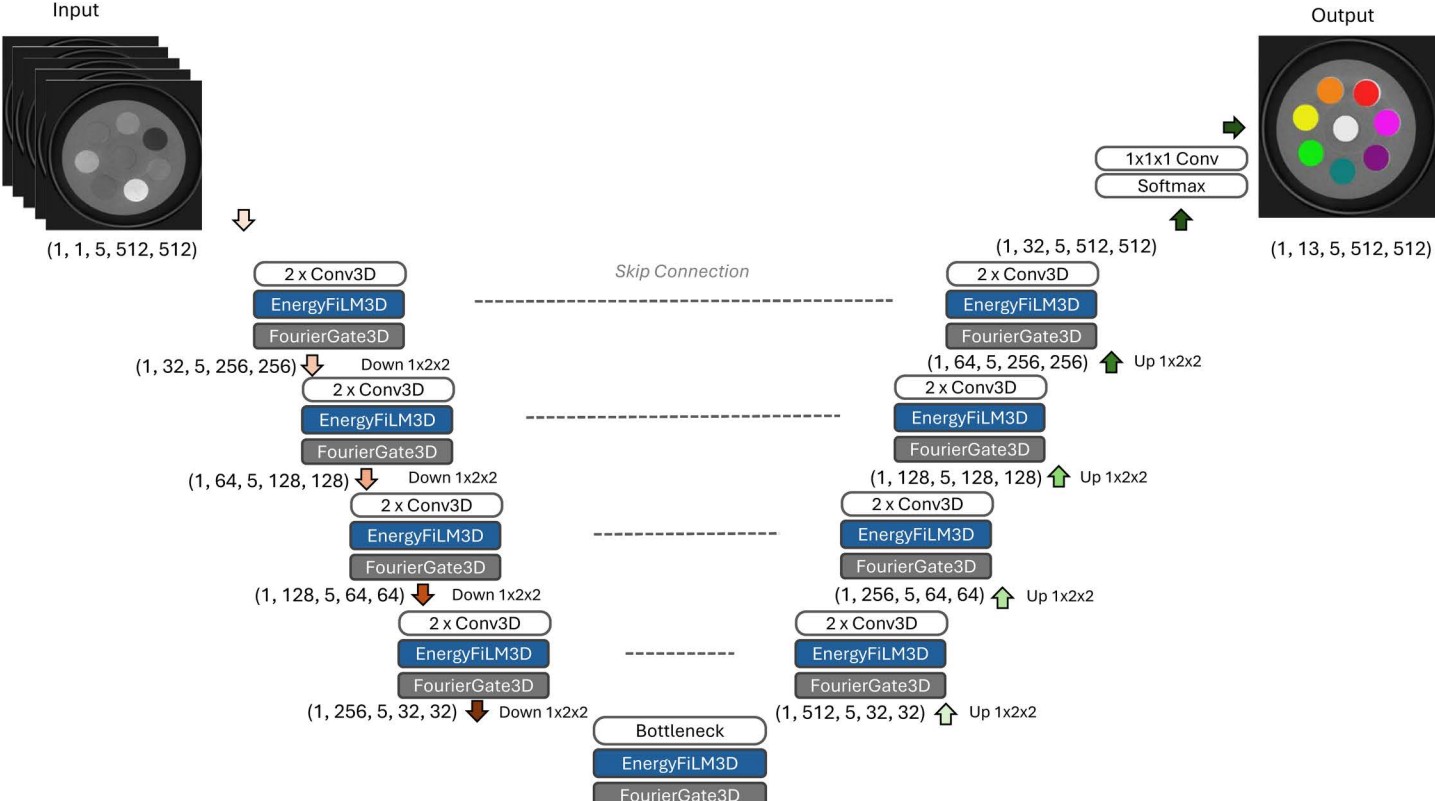

**Fig 3. Architecture of the proposed SPFF–UNet.** A four-level 3D UNet preserves the spectral axis by using anisotropic down/upsampling (no pooling along energy *F*). Two custom modules—EnergyFiLM (per-energy affine modulation) and FourierGate (rFFT-based spectral gating)—are inserted after each double-convolution block in the encoder, bottleneck, and decoder, while skip connections retain full spectral resolution. At the network output, the preserved spectral information is fused to produce 13 class logits per spatial voxel, followed by softmax to generate dense voxel-wise labels. Inputs are SPCCT volumes with five energy bins; outputs are dense voxel-level labels for 13 material classes (hydroxyapatite and iodine concentrations, soft-tissue surrogates). SPCCT: spectral photon-counting CT.

Training uses a composite objective with multiclass cross-entropy and a macro-averaged Dice term computed over foreground classes to mitigate imbalance [49]:

$$\mathcal{L} = \mathcal{L}_{\mathrm{CE}}(\hat{\mathbf{p}}, \mathbf{g}) \; + \; \left(1 - \overline{\mathrm{Dice}}(\hat{\mathbf{p}}, \mathbf{g})\right) \tag{1}$$

$$\overline{Dice}(\hat{\mathbf{p}}, \mathbf{g}) = \frac{2 \sum_{c \in \mathcal{C}_{fg}} \sum \hat{\mathbf{p}}_c \, \mathbf{g}_c}{\sum_{c \in \mathcal{C}_{fg}} \sum \hat{\mathbf{p}}_c \; + \; \sum_{c \in \mathcal{C}_{fg}} \sum \mathbf{g}_c \; + \; \varepsilon} \tag{2}$$

$$\mathcal{L}_{CE}(\hat{\mathbf{p}}, \mathbf{g}) = -\frac{1}{N} \sum_{i=1}^{N} \sum_{c=1}^{13} g_{c,i} \, \log \hat{p}_{c,i} \tag{3}$$

Here, class probabilities $\hat{\mathbf{p}}$ = softmax(logits), $\mathbf{g}$ is ground truth, $\mathcal{C}_{fg}$ is the set of foreground classes (background excluded), $\varepsilon = 10^{-6}$, and $N = B \times H \times W$ is the number of output voxels per batch. Unless stated otherwise, we use base channels 32, InstanceNorm3d, and LeakyReLU (negative slope 0.01). Optimization uses Adam (learning rate $1 \times 10^{-4}$, no weight decay) with ReduceLROnPlateau (factor 0.5, patience 5) monitored on validation macro Dice.

To quantify each module's contribution under identical spectral-preserving sampling, we evaluate four controlled variants of the full **SPFF–UNet** (EnergyFiLM + FourierGate + spec-SE). All variants use 1×2×2 down-/upsampling, with all other settings fixed and background excluded from metrics. These variants are: (a) **Plain-UNet** — no spec-SE, no EnergyFiLM, no FourierGate; (b) **SP-UNet** — spec-SE only; (c) **E-SP-UNet** — spec-SE + EnergyFiLM (no FourierGate); (d) **FG-SP-UNet** — spec-SE + FourierGate (no EnergyFiLM). Results are reported as mean ± SD over three seeds.

## Evaluation protocol

All six models (3D UNet, ResUNet++, R2UNet3D, UNETR, Swin UNETR, SPFF–UNet) were evaluated on a held-out external test scan (the fifth phantom, unseen during training) to assess generalization to a new spectral configuration. Model complexity metrics—including parameter count, multiply–accumulate operations (MACs), and floating point operations (FLOPs)—were computed using consistent input dimensions and batch sizes via the ptflops Python package. These computational metrics are reported for SPFF–UNet and comparators in the Discussion section. Voxel-level probabilities were obtained with a class-wise softmax; hard labels were the $\arg\max$ over classes. We report mean±SD across three independent seeds. Evaluation comprised (i) quantitative analysis—macro overall and per-class performance—and (ii) qualitative analysis with voxel-level overlays and slice-wise Dice-error plots.

## Overall performance

For each class $c$ and test slice, we compute voxel-level counts of true positives (TP), false positives (FP), false negatives (FN), and true negatives (TN). Metrics are defined as follows:

$$Dice_c = \frac{2\,TP}{2\,TP + FP + FN} \tag{4}$$

$$Sensitivity_c \; (Recall_c) = \frac{TP}{TP + FN} \tag{5}$$

$$Specificity_c = \frac{TN}{TN + FP} \tag{6}$$

$$Precision_c = \frac{TP}{TP + FP} \tag{7}$$

$$IoU_c = \frac{TP}{TP + FP + FN} \qquad (8)$$

If a slice contains no positives for class $c$, Dice and Sensitivity/Recall for that slice are treated as NaN and omitted from means. Macro (overall) scores are the unweighted mean over foreground classes (background excluded), using NaN-robust averaging. In this work, background denotes any voxel not belonging to the 12 target classes, including air, phantom housing material, and other unlabeled regions. Results are reported as mean±SD across three seeds to assess reproducibility.

### Per-class performance

Per-class metrics were computed per slice and averaged across slices. The primary per-class metric is Dice in the main text; Sensitivity/Recall, Specificity, Precision, and IoU are provided in the Supplementary. If a class was absent in a slice (no positive ground truth), its Dice and Sensitivity/Recall were recorded as NaN and omitted from that slice's averages. Classes absent from the external test set are indicated in figure legends and excluded from summaries. Results are mean±SD over three seeds to assess reproducibility.

### Qualitative and slice-wise error analyses

We generated voxel-level overlays on representative slices for all six models to assess boundary adherence, label swaps, and spurious predictions. To examine stability beyond averages, we plotted per-slice Dice error $e_c = 1 - Dice_c$ for hydroxy-apatite (HA800/HA400/HA200/HA100) and iodine (I15/I10/I5) across models, showing the mean (solid line) and mean ± 1.96 SD (shaded band) as a 95% reference range. This visualization is mean–difference–style for interpretability and is not a formal Bland–Altman analysis (the reference Dice is fixed at 1).

## Results and discussion

We report both quantitative and qualitative findings to evaluate segmentation performance across models. Quantitative results comprise overall (macro-averaged) performance and per-class performance; qualitative results include voxel-level overlays and slice-wise error analyses.

For readability, models are presented in a fixed order: the canonical baseline 3D UNet first; then other published baselines sorted by macro-averaged Dice (ascending)—UNETR, R2UNet3D, Swin UNETR—with the strongest published baseline (ResUNet++) next; and finally our proposed SPFF–UNet.

### Overall performance

We present SPFF–UNet, a spectral-preserving 3D UNet that integrates spectral squeeze-excitation, EnergyFiLM, and FourierGate to predict discrete, concentration-aware voxel labels directly from multi-energy SPCCT volumes. Table 2 summarizes the macro-averaged segmentation metrics on the external test set, reporting mean± SD over three seeds with background excluded. The table allows direct comparison across the six models, highlighting SPFF–UNet's improvements in overall performance and robustness over baseline models.

SPFF–UNet achieved the highest performance (Dice 0.72, IoU 0.59, sensitivity 0.73, precision 0.71), surpassing the canonical 3D UNet by +0.24 Dice, +0.32 IoU, +0.17 sensitivity, and +0.26 precision, and the strongest published baseline, ResUNet++, by +0.06 Dice, +0.13 IoU, +0.06 sensitivity, and +0.10 precision; specificity was near ceiling (0.97–1.00).

### Per-class performance

Fig 4 shows per-class Dice (12 foreground classes) for each model, computed on each test slice and averaged within class (mean±SD over three seeds). Background is displayed for reference only and is excluded from macro averages elsewhere.

**Table 2. Macro-averaged segmentation performance on the external test set.** Metrics are computed over foreground classes only (background excluded; background = all voxels not belonging to the 12 target classes). Values are mean ± SD across three seeds. IoU = Intersection-over-Union. The first five rows are baseline models; SPFF–UNet is the proposed model.

| Model | Dice | Sensitivity | Specificity | Precision | IoU |
|---|---|---|---|---|---|
| 3DUNet | 0.48 ± 0.07 | 0.56 ± 0.07 | 1.00 ± 0.00 | 0.45 ± 0.07 | 0.27 ± 0.04 |
| UNETR | 0.44 ± 0.02 | 0.43 ± 0.03 | 1.00 ± 0.00 | 0.44 ± 0.03 | 0.28 ± 0.02 |
| R2UNet3D | 0.57 ± 0.01 | 0.62 ± 0.01 | 0.97 ± 0.00 | 0.47 ± 0.02 | 0.35 ± 0.01 |
| SwinUNETR | 0.64 ± 0.02 | 0.65 ± 0.02 | 1.00 ± 0.00 | 0.54 ± 0.02 | 0.42 ± 0.02 |
| ResUNet++ | 0.66 ± 0.02 | 0.67 ± 0.02 | 1.00 ± 0.00 | 0.61 ± 0.03 | 0.46 ± 0.02 |
| SPFF–UNet | 0.72 ± 0.01 | 0.73 ± 0.01 | 1.00 ± 0.00 | 0.71 ± 0.01 | 0.59 ± 0.01 |

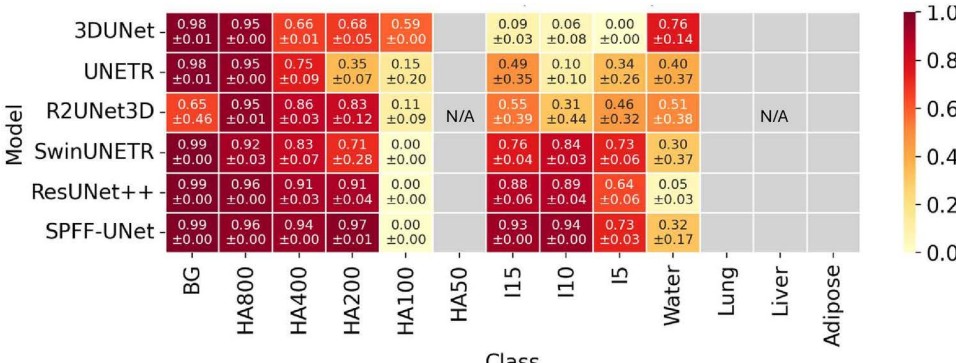

**Fig 4. Per-class Dice heatmaps for six models (five published baselines and the proposed SPFF–UNet).** Each cell shows mean ± SD across three seeds on the external test set. Background (BG; all voxels not belonging to the 12 target classes) is shown for reference only. The 12 target classes are HA800, HA400, HA200, HA100, HA50, I15, I10, I5, water, adipose, liver, and lung; cell color encodes the per-class Dice mean. Rows correspond to models and columns to classes. Cells labeled N/A (light gray) indicate classes absent from the external test but present in the training set, and are excluded from all summary statistics. A consistent colormap and class order are used across panels. HA: hydroxyapatite; I: iodine.

Per-class Dice scores reveal that for HA, performance at the highest concentration is already saturated (HA800 is ≈ 0.95 − 0.96 for all models), and SPFF–UNet offers little additional benefit. In contrast, mid/low-contrast HA rods show large improvements. For HA400, Dice increases from 0.66 with 3D UNet to 0.91 with ResUNet++ and 0.94 with SPFF–UNet. For HA200, the pattern is similar: 0.68 (3D UNet), 0.91 (ResUNet++), and 0.97 (SPFF–UNet). Near the detection limit (HA100), all methods struggle: 3D UNet still achieves 0.59, whereas the more advanced baselines and SPFF–UNet yield values between 0 and 0.15, suggesting that this concentration is effectively below the reliable sensitivity of our pipeline. For iodine, SPFF–UNet yields the most significant improvements at low concentrations. Dice for I15, I10, and I5 reach 0.93, 0.94, and 0.73, respectively, compared with 0.09, 0.06, and 0.00 for 3D UNet. Even the strongest published baseline, ResUNet++, attains lower scores (0.88, 0.89, and 0.64), so SPFF–UNet still outperforms it by about 0.05–0.09 Dice. Water remains challenging and variable across models: SPFF–UNet achieves a Dice of 0.32, similar to the mid-tier baselines (0.30–0.51) but clearly lower than 3D UNet (0.76), indicating a trade-off between optimising HA/iodine contrast and background water segmentation. Overall, the largest benefits of SPFF–UNet concentrate in mid/low-contrast HA (HA400–HA200) and low-concentration iodine (I15/I10/I5), with minimal added value at saturated HA800 and persistently limited performance for HA100 and water.

Class-wise means for Sensitivity (Recall), Precision, and IoU are provided in Supplementary S1 Fig.

## Qualitative and slice-wise error analyses

Fig 5 shows side-by-side voxel-level overlays from all six models on the same representative slices of the external test scan. These overlays are illustrative only and do not affect the quantitative results reported elsewhere.

Qualitative examples mirror these trends. 3D UNet recovers most HA but frequently mislabels iodine inserts as HA. UNETR recovers HA800-HA200 yet under-segments water and HA100, with iodine partly under-segmented and confused with adjacent HA bins (e.g., I15 with HA400, I10 with HA200, I5 with HA100). R2UNet3D shows boundary leakage and mixing among spectrally proximate classes; iodine inserts are largely under-segmented and mislabeled as HA, and water is often missed. Swin UNETR detects HA and iodine; however, its segmentations exhibit fragmentation and speckle artifacts across several inserts. ResUNet++ more closely follows the reference with steadier boundaries, yet residual mixing and small mislabeled regions persist in I5, water, and HA100. SPFF–UNet most closely approximates the ground truth: insert shapes and labels are preserved with minimal edge artifacts, cross-contamination between iodine and HA is reduced, and background suppression is clean; remaining deviations occur mainly in water (partial confusion with HA100) and in HA100 near its detection limit. Panels depict representative slices; quantitative results are summarized in Table 2.

Fig 6 compares the baseline 3D UNet, the highest-performing Dice-based published baseline (ResUNet++), and the proposed SPFF–UNet for per-slice segmentation error in HA and iodine, providing insights beyond macro averages.

For hydroxyapatite classes (top row, (A)–(C)), all models yield similar mean errors, but the baseline 3D UNet (A) exhibits the widest variability and most pronounced outliers. ResUNet++ (B) demonstrates moderately reduced dispersion, while SPFF–UNet (C) achieves the tightest ± 1.96 SD bounds and the lowest mean error, indicating

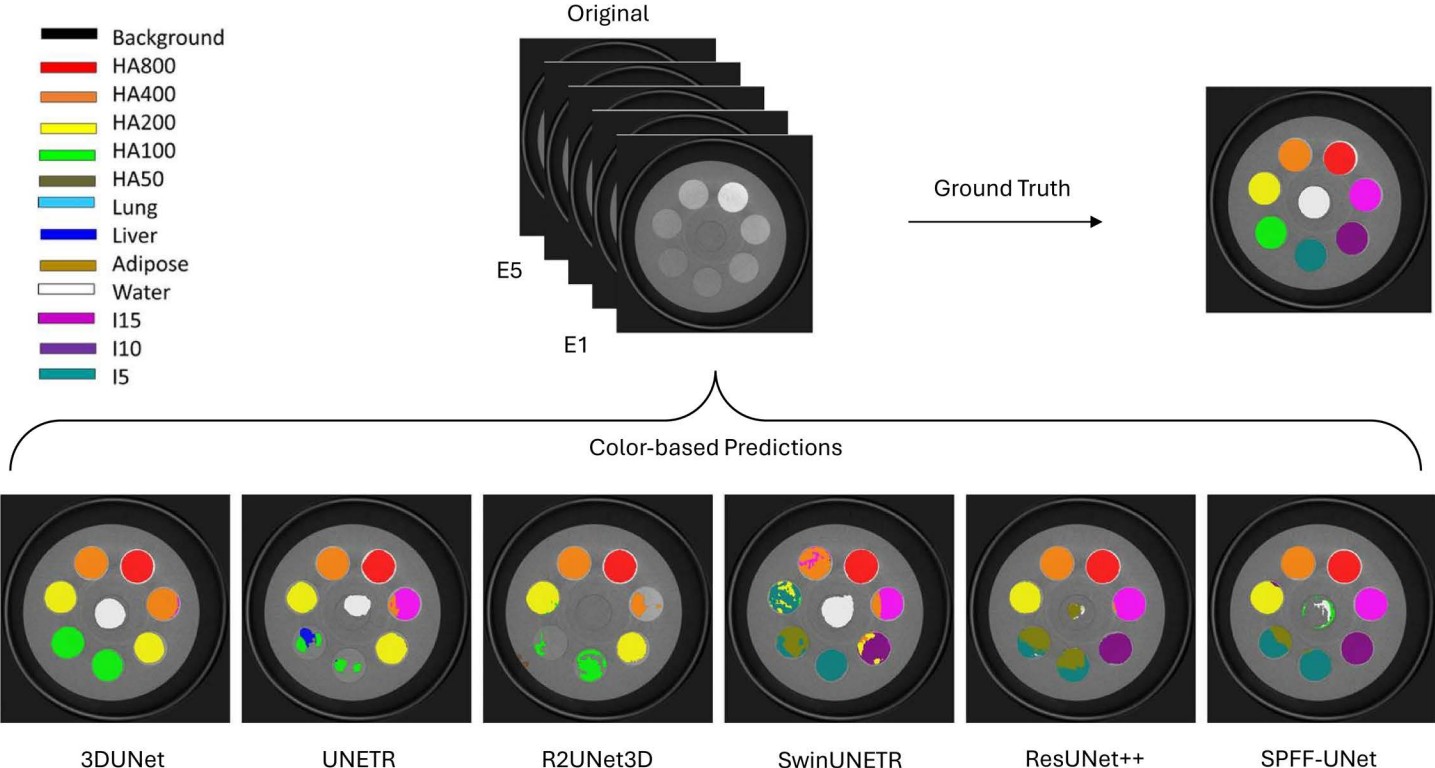

**Fig 5. Qualitative overlays of voxel-level segmentations on the external test scan.** Columns (left to right): 3D UNet, UNETR, R2UNet3D, Swin UNETR, ResUNet++, and SPFF–UNet (proposed). Predicted labels are argmax maps color-coded on grayscale SPCCT slices; all panels use identical windowing and a shared class colormap/legend. HA: hydroxyapatite; I: iodine. E: Energy bin (7-12, 12-15, 15-18, 18-21, 21-120 keV).

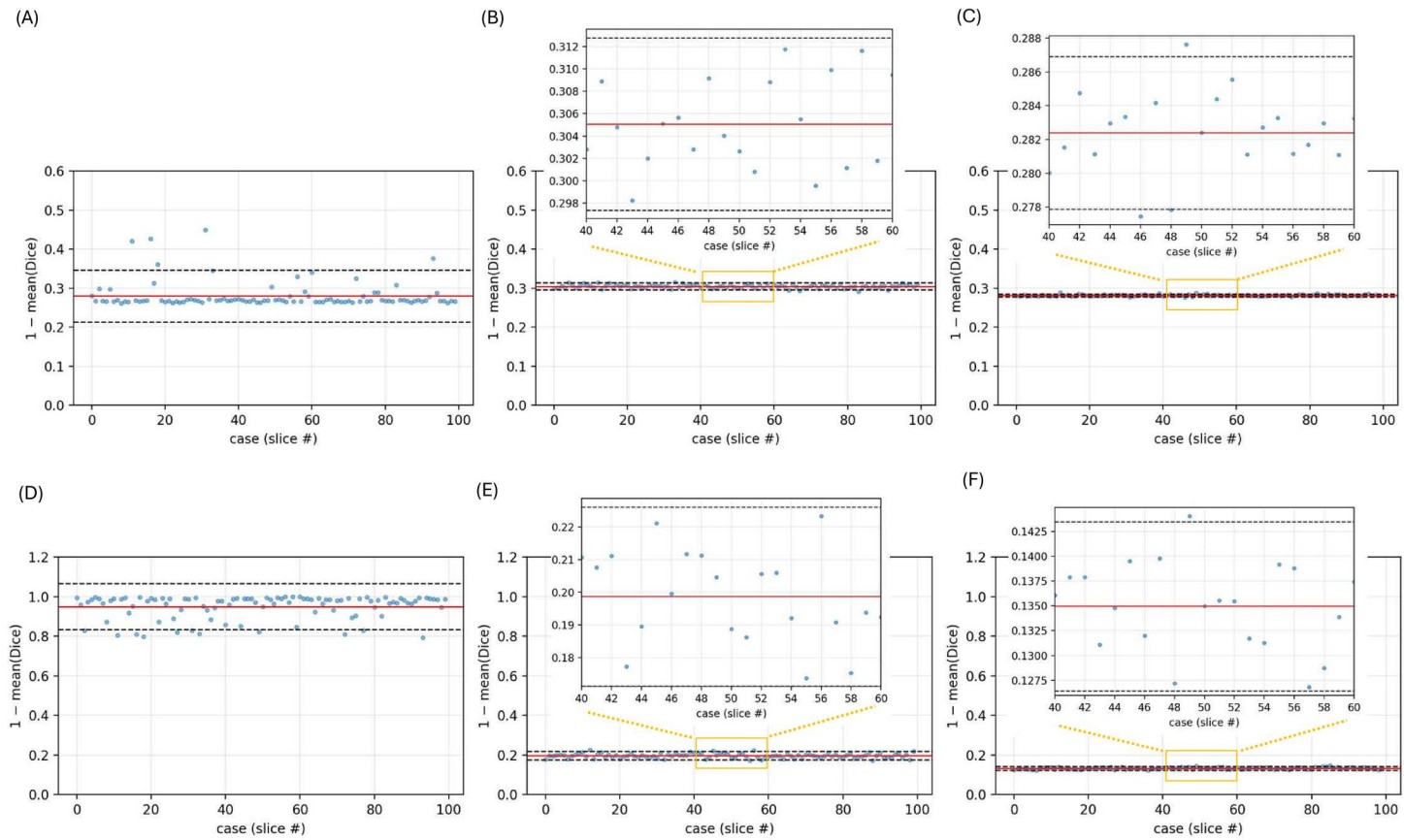

**Fig 6. Slice-wise Dice error on the external test scan for three comparators. (A), (D):** baseline 3D UNet; **(B), (E):** highest-Dice published baseline (ResUNet++); **(C), (F):** proposed SPFF–UNet. Top row **(A)–(C)** corresponds to the average of hydroxyapatite classes (HA800, HA400, HA200, HA100); bottom row **(D)–(F)** corresponds to the average of iodine classes (I15, I10, I5). For each model, per-slice errors are plotted with a solid mean line and dashed ±1.96 SD band, averaged over three seeds. The *x*-axis represents the test slice index; the *y*-axis shows the error, defined as $e = 1 - \text{Dice}$. Insets provide zoomed views for ResUNet++ and SPFF–UNet in the hydroxyapatite row, and for SPFF–UNet in the iodine row. These plots show per-slice deviation from perfect overlap (Dice = 1), offering a diagnostic view of segmentation consistency across slices, not classical Bland-Altman plots.

more consistent voxel-level segmentation across slices. For iodine classes (bottom row, (D)–(F)), performance differences are more marked: 3D UNet (D) shows the highest error and greatest variance, ResUNet++ (E) improves moderately, and SPFF–UNet (F) again attains the best overall stability, with the lowest mean error and narrowest confidence bands. These trends support the improved slice-wise consistency of SPFF–UNet across both material types.

In addition to improved accuracy, SPFF–UNet remains relatively compact, comprising 5.49 million parameters and requiring 560.65 G MACs (approximately 1121.3 G FLOPs), compared with 22.58 million parameters and 1068.41 G MACs (≈2136.8 G FLOPs) for 3D UNet and 16.64 million parameters and 285.94 G MACs (≈571.9 G FLOPs) for ResUNet++. Although its MAC count is higher than that of ResUNet++, SPFF–UNet uses substantially fewer parameters than several comparators while achieving the best segmentation performance, indicating a favorable accuracy–complexity trade-off. These properties suggest that SPFF–UNet may offer a favorable accuracy–complexity balance for future SPCCT implementations, although deployment-oriented runtime validation remains necessary.

## Ablation study of the proposed model

To better understand the contribution of each component within SPFF–UNet, we conducted an ablation study as indicated in Table 3. It reports macro-averaged metrics on four controlled spectral-preserving variants of the full SPFF–UNet, evaluated on the external test set, highlighting how each spectral component affects overlap and precision.

Adding spectral squeeze-and-excitation to the plain backbone (SP-UNet vs. Plain-UNet) yields +0.074 Dice, +0.071 sensitivity, +0.072 precision, and +0.104 IoU, supporting channel reweighting once the spectral dimension is preserved (1×2×2). Adding EnergyFiLM alone (E-SP-UNet vs. SP-UNet) trades small Dice/sensitivity for calibration gains (−0.017 Dice, −0.021 sensitivity; +0.030 precision, +0.011 IoU), consistent with fewer false positives. Adding FourierGate alone (FG-SP-UNet vs. SP-UNet) increases Dice slightly (+0.002) and changes sensitivity minimally (−0.002), but yields a larger drop in precision (−0.035) and IoU (−0.026), reflecting a detection–precision trade-off. The full model (SPFF–UNet) balances these effects: vs. SP-UNet it keeps Dice similar ($\Delta \approx 0$) with a slight sensitivity drop (−0.008) but gains +0.081 precision (0.710) and +0.061 IoU (0.587); vs. FG-SP-UNet it improves precision/IoU by +0.116/+0.087 with essentially unchanged Dice (−0.002) and a small sensitivity change (−0.006). This occurs because Dice balances precision and sensitivity; when sensitivity remains stable but precision improves—as in this case—the Dice coefficient can appear unchanged even though false positives are reduced. IoU, which penalizes false positives more directly, reflects this improvement more strongly. Finally, vs. E-SP-UNet it improves all four metrics (+0.017 Dice, +0.013 sensitivity, +0.051 precision, +0.050 IoU). Against the plain backbone, SPFF–UNet provides the largest lift (+0.074 Dice, +0.063 sensitivity, +0.153 precision, +0.165 IoU). Specificity is near ceiling for all variants (0.997–0.998) and thus less discriminative; patterns are consistent across three seeds, though some pairwise differences are modest and warrant confirmation on larger datasets.

Although public spectral CT datasets are beginning to emerge, direct external benchmarking remains limited for the present task. For example, the open 2022 AAPM deep-learning spectral CT grand challenge dataset [50,51] provides large-scale low/high-kVp spectral CT data for simulated breast phantoms and tissue-map prediction, which differs substantially from our setting of five-bin SPCCT and voxel-level concentration-aware labeling of hydroxyapatite and iodine. As a result, currently available public datasets are useful for methodological transfer studies, but they do not constitute a task-matched benchmark for the specific material-classification problem addressed here. Future work should therefore evaluate cross-dataset transfer and domain adaptation on publicly available spectral CT resources as well as larger multi-site SPCCT cohorts when such data become available.

This work also has limitations. Firstly, the results are phantom-based and may not fully capture clinical variability, such as patient motion, beam hardening, device heterogeneity, and pathology prevalence. Secondly, the available phantom included only a sparse set of discrete concentrations; we could evaluate the model only at the HA and iodine levels

**Table 3. Ablation on a spectral-preserving UNet backbone.** Macro segmentation metrics (mean ± SD over three seeds) on the external test set. All variants retain the energy axis via 1×2×2 down/upsampling; background is excluded from macro averages. Rows are ordered a→e (plain→full). IoU = Intersection-over-Union.

| Variant | Dice | Sensitivity | Specificity | Precision | IoU |
|---|---|---|---|---|---|
| Plain-UNet[a] | 0.650 ± 0.013 | 0.664 ± 0.011 | 0.997 ± 0.000 | 0.557 ± 0.007 | 0.422 ± 0.009 |
| SP-UNet[b] | 0.724 ± 0.013 | 0.735 ± 0.014 | 0.998 ± 0.000 | 0.629 ± 0.006 | 0.526 ± 0.012 |
| E-SP-UNet[c] | 0.707 ± 0.012 | 0.714 ± 0.013 | 0.997 ± 0.000 | 0.659 ± 0.009 | 0.537 ± 0.012 |
| FG-SP-UNet[d] | 0.726 ± 0.012 | 0.733 ± 0.013 | 0.997 ± 0.000 | 0.594 ± 0.006 | 0.500 ± 0.011 |
| **SPFF–UNet[e]** | 0.724 ± 0.012 | 0.727 ± 0.011 | 0.998 ± 0.000 | 0.710 ± 0.006 | 0.587 ± 0.012 |

[a]Plain UNet backbone: no spec-SE, no EnergyFiLM, no FourierGate. [b] spec-SE only. [c] EnergyFiLM only (spec-SE on). [d] FourierGate only (spec-SE on).
[e] **Full model (SPFF–UNet)**: EnergyFiLM + FourierGate + spec-SE.

present in the rods, and not at intermediate or unseen concentrations (e.g., iodine 12 mg/mL), leaving SPFF–UNet's ability to interpolate between trained concentration levels untested. Thirdly, the held-out external evaluation consisted of a single independent phantom scan; accordingly, the reported mean ± SD across seeds reflects training stochasticity and model robustness under repeated optimization rather than population-level variance across independently sampled phantoms or patients. Finally, some pairwise differences are modest and therefore warrant confirmation on larger, multi-center datasets with diverse scanners and protocols.

## Conclusion

We emphasize that this is a phantom-based proof-of-concept study and that no near-term clinical claims are made; in vivo validation across scanners, protocols, and patient populations will be required before any clinical translation can be considered. We introduce SPFF–UNet, a spectral-preserving 3D UNet with EnergyFiLM and FourierGate for concentration-aware voxel labeling on SPCCT. On a held-out phantom, SPFF–UNet outperformed published baselines, with clear gains for mid/low-contrast HA (HA400–HA200) and low-concentration iodine (I15/I10/I5); HA800 was saturated and HA100 remained near the detection limit. Ablations indicate complementary effects: spectral squeeze-excitation yielded the largest Dice/sensitivity gains; EnergyFiLM improved precision; FourierGate increased Dice but reduced precision, reflecting a detection–precision trade-off. In combination, these modules balanced the trade-off, yielding higher precision without loss of Dice. If validated in vivo across scanners and protocols, concentration-aware HA segmentation could support quantitative assessment of calcium burden in crystal-induced arthropathies, while improved iodine–HA separation may help reduce misclassification on contrast-enhanced SPCCT examinations. These findings motivate future prospective studies to assess generalizability and clinical utility.

## Supporting information

**S1 Fig. Per-class heatmaps of all models.** (A) Sensitivity. (B) Precision. (C) IoU. Rows are models; columns are classes. Values are mean ± SD across three seeds on the external test. Light–gray cells labeled N/A indicate classes absent from the ground truth and excluded from summaries; for these, Precision/IoU are undefined when no positives are predicted (TP = 0, FP = 0). For the absent soft–tissue classes, SPFF–UNet is consistently N/A, indicating it did not hallucinate labels. Several baselines instead show 0.00 ± 0.00 in the same columns—this occurs when TP = 0 but FP > 0, collapsing Precision and IoU to zero because false positives were produced for classes that should be empty. This behavior is consistent with our results.BG: Background.
(TIF)

## Author contributions

**Conceptualization:** Nadine Francis.

**Data curation:** Nadine Francis.

**Formal analysis:** Nadine Francis.

**Funding acquisition:** Aamir Raja.

**Investigation:** Nadine Francis.

**Methodology:** Nadine Francis.

**Project administration:** Aamir Raja.

**Resources:** Aamir Raja.

**Software:** Nadine Francis.

**Supervision:** Mohamed L Seghier, Nabil Maalej, Aamir Raja.

**Validation:** Nadine Francis, Mohamed L Seghier.

**Visualization:** Nadine Francis.

**Writing – original draft:** Nadine Francis.

**Writing – review & editing:** Mohamed L Seghier, Nabil Maalej, Aamir Raja.

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
