## [Decision Letter · Decision Letter 0]

24 Feb 2026

PONE-D-25-63592Voxel-wise deep learning segmentation of hydroxyapatite and iodine in spectral photon-counting CT: A quantitative phantom studyPLOS One

Dear Dr. Francis,

Thank you for submitting your manuscript to PLOS ONE. After careful consideration, we feel that it has merit but does not fully meet PLOS ONE’s publication criteria as it currently stands. Therefore, we invite you to submit a revised version of the manuscript that addresses the points raised during the review process.

We look forward to receiving your revised manuscript.

Kind regards,

Abdel-Razzak M. Al-Hinnawi, Ph.D.

Academic Editor

PLOS One

Journal Requirements:

https://journals.plos.org/plosone/s/file?id=wjVg/PLOSOne_formatting_sample_main_body.pdf and and and and https://journals.plos.org/plosone/s/file?id=ba62/PLOSOne_formatting_sample_title_authors_affiliations.pdf

2. Please note that PLOS One has specific guidelines on code sharing for submissions in which author-generated code underpins the findings in the manuscript. In these cases, all author-generated code must be made available without restrictions upon publication of the work. Please review our guidelines at https://journals.plos.org/plosone/s/materials-and-software-sharing#loc-sharing-code and ensure that your code is shared in a way that follows best practice and facilitates reproducibility and reuse.

3. In the online submission form, you indicated that All relevant data will be provided upon request.

This project was funded by the Research and Innovation Grant, Khalifa University of Science and Technology, Abu Dhabi, UAE under account number 8474000563.

5. We note that Figure(s) 1, 2, 3, 4, 5, S1 in your submission contain copyrighted images. All PLOS content is published under the Creative Commons Attribution License (CC BY 4.0), which means that the manuscript, images, and Supporting Information files will be freely available online, and any third party is permitted to access, download, copy, distribute, and use these materials in any way, even commercially, with proper attribution. For more information, see our copyright guidelines: http://journals.plos.org/plosone/s/licenses-and-copyright.

a. You may seek permission from the original copyright holder of Figure(s) 1, 2, 3, 4, 5, S1 to publish the content specifically under the CC BY 4.0 license.

6. We notice that your supplementary figures are uploaded with the file type 'Figure'. Please amend the file type to 'Supporting Information'. Please ensure that each Supporting Information file has a legend listed in the manuscript after the references list.

Additional Editor Comments:

1-) Please shorten the abstract. Try to make it more concise. It is not intelleigibly presented. It exceeds the world limit. I think authors can shorten it and put it into a more "concise" and "Fluent" structure.

2-) A concern by a reviewer is that if there are public datasets, how could the findings from this manuscript be generalized to public datasets? A short paragraph in the discussion section is recommended. You may add references in this context or use teh available references.

Reviewers' comments:

Reviewer's Responses to Questions

**Comments to the Author**

1. Is the manuscript technically sound, and do the data support the conclusions?

Reviewer #1: Yes

Reviewer #2: Partly

2. Has the statistical analysis been performed appropriately and rigorously? 

Reviewer #1: Yes

Reviewer #2: Yes

3. Have the authors made all data underlying the findings in their manuscript fully available?

The PLOS Data policy requires authors to make all data underlying the findings described in their manuscript fully available without restriction, with rare exception (please refer to the Data Availability Statement in the manuscript PDF file). The data should be provided as part of the manuscript or its supporting information, or deposited to a public repository. For example, in addition to summary statistics, the data points behind means, medians and variance measures should be available. If there are restrictions on publicly sharing data—e.g. participant privacy or use of data from a third party—those must be specified.requires authors to make all data underlying the findings described in their manuscript fully available without restriction, with rare exception (please refer to the Data Availability Statement in the manuscript PDF file). The data should be provided as part of the manuscript or its supporting information, or deposited to a public repository. For example, in addition to summary statistics, the data points behind means, medians and variance measures should be available. If there are restrictions on publicly sharing data—e.g. participant privacy or use of data from a third party—those must be specified.requires authors to make all data underlying the findings described in their manuscript fully available without restriction, with rare exception (please refer to the Data Availability Statement in the manuscript PDF file). The data should be provided as part of the manuscript or its supporting information, or deposited to a public repository. For example, in addition to summary statistics, the data points behind means, medians and variance measures should be available. If there are restrictions on publicly sharing data—e.g. participant privacy or use of data from a third party—those must be specified.requires authors to make all data underlying the findings described in their manuscript fully available without restriction, with rare exception (please refer to the Data Availability Statement in the manuscript PDF file). The data should be provided as part of the manuscript or its supporting information, or deposited to a public repository. For example, in addition to summary statistics, the data points behind means, medians and variance measures should be available. If there are restrictions on publicly sharing data—e.g. participant privacy or use of data from a third party—those must be specified.

Reviewer #1: No

Reviewer #2: Yes

4. Is the manuscript presented in an intelligible fashion and written in standard English?

Reviewer #1: Yes

Reviewer #2: Yes

5. Review Comments to the Author

Reviewer #1: This manuscript presents a technically sound and carefully executed phantom study investigating voxel-wise, concentration-aware classification of hydroxyapatite and iodine using spectral photon-counting CT and deep learning. The authors propose a spectral-preserving 3D UNet variant (SPFF–UNet) and benchmark it against multiple established architectures under a unified training and evaluation protocol. However, points for improvement / clarification

1) Data availability and reproducibility

To fully align with PLOS ONE’s data-sharing policy, the authors should include a clear Data Availability Statement specifying whether datasets, trained models, or code will be made publicly available, and where.

2) Clinical claims and framing

While the discussion is generally cautious, a few statements could be further softened to avoid any implication of near-term clinical applicability. Explicitly reiterating that no clinical claims are made and that in vivo validation is required would strengthen alignment with the journal’s scope.

3) Justification of classification vs. regression

Although implicit in the motivation, a brief explicit discussion explaining why discrete concentration-aware classification was favored over continuous regression (commonly used in spectral CT) would improve clarity for a broad readership.

4) Statistical generalizability

The authors may wish to emphasize that the external test set consists of a single phantom scan and that reported variability reflects training stochasticity rather than population-level variance. This is a limitation rather than a flaw, but making it explicit would preempt reviewer concerns.

Reviewer #2: must be compare the used dataset with global used dataset to be a strong reference and a good scientific methodology used and continue compare with another standard data set from the outside to be more accuracy and confidential

6. PLOS authors have the option to publish the peer review history of their article (what does this mean?). If published, this will include your full peer review and any attached files.). If published, this will include your full peer review and any attached files.). If published, this will include your full peer review and any attached files.). If published, this will include your full peer review and any attached files.

...

Reviewer #1: No

Reviewer #2: **Yes:** Dr Mohammad Abosaaleek isra universityDr Mohammad Abosaaleek isra universityDr Mohammad Abosaaleek isra universityDr Mohammad Abosaaleek isra university

---

## [Author Response · Author response to Decision Letter 1]

2 Mar 2026

Dear Editor and Reviewers,

We sincerely thank you for the careful evaluation of our manuscript and for the constructive comments and additional journal requirements. We greatly appreciate the time and effort invested in the review process. We have carefully considered all comments and revised the manuscript accordingly. Below we provide a point-by-point response to all items raised. All changes in the revised manuscript are highlighted in red for ease of reference.

Editor Comments

1. Please shorten the abstract. Try to make it more concise. It is not intelligibly presented. It exceeds the word limit.

Response: Thank you for this helpful suggestion. We substantially shortened and rewrote the abstract to improve clarity, fluency, and concision while preserving the study objective, core methodology, principal findings, and main conclusion.

2. A concern by a reviewer is that if there are public datasets, how could the findings from this manuscript be generalized to public datasets? A short paragraph in the discussion section is recommended.

Response: We agree that generalizability beyond the present phantom dataset should be discussed more explicitly. We added a paragraph to the Discussion clarifying that directly matched public benchmarks for five-bin SPCCT hydroxyapatite/iodine voxel-wise labeling are currently limited. We also note that related public spectral CT datasets exist (e.g., the 2022 AAPM deep-learning spectral CT grand challenge dataset~\cite{sidky2024aapm_dlspectralct_dataset,sidky2024aapm_report}), but these differ substantially in acquisition type, task definition, and target labels, and therefore do not provide a like-for-like external benchmark for the present study. We highlight this as an important direction for future work.

Reviewer #1

1. Data availability and reproducibility

To fully align with PLOS ONE’s data-sharing policy, the authors should include a clear Data Availability Statement specifying whether datasets, trained models, or code will be made publicly available, and where.

Response: Thank you for this important suggestion. We added a clear Data Availability Statement to the manuscript. The dataset used in this study is publicly available in IEEE DataPort (DOI: 10.21227/gbhn-nk95). The full code used for model training, evaluation, and figure generation has been made publicly available and archived on Zenodo (v1.0.0 DOI: 10.5281/zenodo.18797577; all versions DOI: 10.5281/zenodo.18797576) and is mirrored on GitHub (https://github.com/NF-91/spff-unet-spcct). Source data underlying the reported tables and figures are provided within the manuscript and its Supporting Information files.

2. Clinical claims and framing

While the discussion is generally cautious, a few statements could be further softened to avoid any implication of near-term clinical applicability. Explicitly reiterating that no clinical claims are made and that in vivo validation is required would strengthen alignment with the journal’s scope.

Response: We appreciate this suggestion and agree that the phantom-based nature of the study should remain central to the framing. We revised the Discussion and Conclusion to further soften statements that could be interpreted as implying near-term clinical applicability. We now explicitly reiterate that no clinical claims are made and that in vivo and multi-center validation will be required before any clinical translation can be considered.

3. Justification of classification vs. regression

Although implicit in the motivation, a brief explicit discussion explaining why discrete concentration-aware classification was favored over continuous regression (commonly used in spectral CT) would improve clarity for a broad readership.

Response: We thank the reviewer for this important point. We added a brief discussion clarifying why a classification formulation was used in this work rather than continuous regression. In brief, our goal was to generate directly interpretable voxel-wise concentration labels for predefined hydroxyapatite and iodine classes, whereas regression would require an additional post hoc discretization step for the specific decision context considered here.

4. Statistical generalizability

The authors may wish to emphasize that the external test set consists of a single phantom scan and that reported variability reflects training stochasticity rather than population-level variance.

Response: We agree and clarified this limitation explicitly. The revised manuscript states that the external test set comprises a single independent phantom scan and that the reported mean ± SD across seeds reflects variability due to repeated training (training stochasticity/model robustness), rather than population-level statistical uncertainty across independently sampled phantoms or patients.

Reviewer #2

Comment: The used dataset should be compared with globally used datasets to provide a stronger reference and a more robust scientific methodology, and should also be compared with another standard outside dataset to improve accuracy and confidence.

Response: We appreciate the reviewer’s emphasis on external reference datasets and broader generalizability. We added a paragraph to the Discussion addressing this point. Specifically, we clarify that directly comparable public datasets for five-bin SPCCT voxel-wise hydroxyapatite/iodine concentration labeling are currently limited. We also discuss related public spectral CT resources (e.g., the 2022 AAPM deep-learning spectral CT grand challenge dataset~\cite{sidky2024aapm_dlspectralct_dataset,sidky2024aapm_report}) and explain why they are not directly task-matched to the present phantom study. This limitation is now acknowledged more explicitly, and future validation on larger external and public datasets is highlighted as an important next step.

---

## [Editor Report · Decision Letter 1]

24 Mar 2026

Voxel-wise deep learning segmentation of hydroxyapatite and iodine in spectral photon-counting CT: A quantitative phantom study

PONE-D-25-63592R1

Dear Dr. Francis,

We’re pleased to inform you that your manuscript has been judged scientifically suitable for publication and will be formally accepted for publication once it meets all outstanding technical requirements.

Kind regards,

Abdel-Razzak M. Al-Hinnawi, Ph.D.

Academic Editor

PLOS One

Additional Editor Comments :

Typo recommendation:

Please, shift the statement in lines 398-399-400 (first sentence in the discussion) to the end of the conclusion section : "This is a phantom-based proof-of-concept study and that no near-term clinical claims are made; in vivo validation across scanners, protocols, and patient populations will be required before clinical translation can be considered." (i.e., It is recommended that this statement to be the last statement in the Discussion section) . Thank you.

---

## [Editor Report · Acceptance letter]

PONE-D-25-63592R1

PLOS One

Dear Dr. Francis,

I'm pleased to inform you that your manuscript has been deemed suitable for publication in PLOS One. Congratulations! Your manuscript is now being handed over to our production team.

Kind regards,

on behalf of

Professor Abdel-Razzak M. Al-Hinnawi

Academic Editor

PLOS One